# Time-Slotted Spreading Factor Hopping for Mitigating Blind Spots in LoRa-Based Networks

**DOI:** 10.3390/s22062253

**Published:** 2022-03-14

**Authors:** Alejandro Iglesias-Rivera, Roald Van Glabbeek, Erik Ortiz Guerra, An Braeken, Kris Steenhaut, Hector Cruz-Enriquez

**Affiliations:** 1Departamento de Electrónica y Telecomunicaciones, Universidad Central “Marta Abreu” de Las Villas, Santa Clara 50100, Villa Clara, Cuba; aiglesas@uclv.cu (A.I.-R.); erik@uclv.edu.cu (E.O.G.); 2Department of Engineering Technology (INDI), Vrije Universiteit Brussel, Pleinlaan 2, 1050 Brussel, Belgium; roald.van.glabbeek@vub.be (R.V.G.); abraeken@vub.be (A.B.); 3Department of Electronics and Informatics (ETRO), Vrije Universiteit Brussel, Pleinlaan 2, 1050 Brussel, Belgium

**Keywords:** IoT, LoRa-based, blind spots, random scheduling

## Abstract

It has been demonstrated that LoRa-based wide area networks (WANs) can cover extended areas under harsh propagation conditions. Traditional LoRaWAN solutions based on single-hop access face important drawbacks related to the presence of blind spots. This paper aims to tackle blind spots and performance issues by using a relaying approach. Many researchers investigating multi-hop solutions consider a fixed spreading factor (SF). This simplifies synchronization and association processes, but does not take advantage of the orthogonality between the virtual channels (i.e., frequency, SF) that help to mitigate blind spots. This paper proposes a time-slotted spreading factor hopping (TSSFH) mechanism that combines virtual channels and time slots into a frame structure. Pseudo-random scheduling is used inside blind spots, which simplifies the end-devices’ communication process and network organization. The results show how collisions decrease inside blind spots when more communication opportunities become available as more relaying nodes can be listening in different cells (i.e., frequency, SF-offset, time-offset). This has a direct impact on the collision-free packet delivery ratio (PDR) metric, which improves when more listening windows are opened, at the expense of faster battery depletion.

## 1. Introduction

The growing interest in the development of Internet of Things (IoT) networks and the increase in the number of connected devices have attracted the attention of researchers and industry. The design of mechanisms to organize massive communication among a large number of IoT devices in an efficient way is still an open issue. LoRaWAN is an open medium access control (MAC) protocol that facilitates the deployment of private networks using unlicensed sub-GHz frequency bands, with low cost implementation because of the hardware’s availability and large areas covered [1].

Some studies have demonstrated how urban environments with high population density can experience spectrum congestion and high levels of interference due to the growing activity in the 868 MHz industrial, scientific, and medical (ISM) band, limiting the potential coverage and capacity of LoRa networks [2,3,4]. However, these possible problems do not completely exclude the use of LoRa in urban areas, as shown in other works [5,6,7].

Radio base stations or gateways (GWs) are located in key places, guaranteeing a wide coverage for all the transmitter nodes belonging to a network. However, it is not always possible to place them in the right location, or to increase their number to avoid blind spots due to factors such as installation cost, locations on private property, contract limitations, management, permissions, planning [8], or temporal channel impairments [7]. A relaying mechanism can improve packet delivery through relay nodes that help so-called disconnected nodes to reach a GW. Relaying can also be useful when nodes lose their direct coverage to a GW, due to GW failures, by leading the packets to alternative GWs.

LoRa networks have features that allow them to operate in a variety of environments. LoRa symbols are modulated using the chirp spread spectrum technique, which can apply different (orthogonal) spreading factors (SFs) corresponding to different data rates (SF7–SF12). Higher SFs offer more robustness to obstacles, interference, and noise by sacrificing energy efficiency and bandwidth for range and reliability [8]. A GW can listen or transmit at any time and at all SFs and frequencies. In the classical LoRaWAN, an ALOHA-like MAC protocol for single-hop, star-based LoRa networks, a common strategy is to start by sending the frame with the lower spreading factor SF7 and, in case of failed acknowledgement reception (due to distance, interference, or collision) increase it to a higher or a random factor. The use of random selection can be a good alternative, but it can be difficult to implement in a relaying or multi-hop scenario as nodes, unlike GWs, can at a given time only transmit or receive on a specific SF and frequency. In a multi-hop network, common time slots could be scheduled or agreed upon among neighbors for successful data communication [9]. Therefore, researchers who address multi-hop LoRa-based MAC protocols simplify their approach by using a fixed spreading factor.

The main motivation for this work was to design and evaluate a relaying mechanism to solve blind spots in LoRa-based wide area networks (WANs). The proposal employs a synchronous MAC protocol that introduces time-slotted spreading factor hopping (TSSFH) to increase transmission opportunities by using multiple SFs. The TSSFH spectrum access method restricts the duty cycle to 1%, a mandatory legal restriction for the uplink communication of LoRaWAN end devices that use the 868 MHz ISM band in Europe. An energy analysis was carried out to show that the TSSFH approach is suitable for power-constrained devices.

The rest of this paper is structured as follows. Related work is presented in Section 2. Section 3 introduces the TSSFH approach. Section 4 presents an energy-related analysis, estimating the impact of the TSSFH approach in the average current draw. The analytic model of the proposed TSSFH is presented in Section 5. Simulations that validate the results regarding the network performance in terms of collision-free packet delivery ratio, overhearing, and idle listening are discussed in Section 6. Section 7 concludes the paper.

## 2. Related Work

In [10] several open research strategies aimed at mitigating LoRa-based WAN limitations are mentioned. These include new channel hopping methods, the use of time division multiple access (TDMA), power reduction for multi-hop solutions, and the combination of ALOHA-based access control with MAC techniques for serving deterministic traffic. Among these are TDMA schedulers that allocate shared resources for ALOHA-based access control and dedicated schedulers for deterministic traffic.

Previous studies, such as [8], addresses important aspects of the network topology and multi-hop forwarding. They focus on applications with sporadic traffic and assume that the loss of a few packets is non-critical. The gateway sends beacons periodically for synchronization. This approach assumes that relay nodes (RN) are not energy-constrained, and future analysis should be performed to assess its energy efficiency. Many authors [8,9] assume that the nodes involved in the multi-hop network know beforehand which role they will play (leaf node, relay node out of GW range, relay node in GW range).

In [11], an exploration of time-slotted medium-access protocols that leverage LoRa at the physical layer are presented. The authors formalize the scheduling, time synchronization, and routing as key challenges in the design of time-slotted solutions. Scheduling is used for resource allocation and can be centralized or distributed. The use of scheduling mechanisms increases the number of overall successful network transmissions at the expense of the protocol overhead Synchronization is essential in time division access protocols and to maintain it, additional transmissions are needed, increasing energy consumption. Meanwhile, time-slotted LoRa communications enable multi-hop and in such scenarios, routing protocols are essential.

Several authors do take advantage of virtual channel (frequency and SF) orthogonality with a TDMA approach inspired by time-slotted channel hopping (TSCH). Examples of this approach are the so-called TSCH-like and TS-LoRa, presented in [12,13], respectively. This research focuses on real-time industrial applications, for which the allocation of transmission opportunities is scheduled by a centralized control. The TSCH-like mechanism is a non-power-constrained approach applied to real-time single-hop scenarios; it does not address the problem of blind spots. It assumes that time slots have a fixed size regardless of the SF used. When transmissions use lower SFs, there is unused time that could otherwise be used to accommodate extra transmissions. A similar approach is presented in [14], in which the authors propose a TSCH adaptation for LoRa operation, ensuring LoRa multi-hop operation using multiple SF, but with each assigned to a different channel, not for the orthogonality and, as such, not to expand the transmission possibilities. In the TS-LoRa proposal, each node uses a simple algorithm to determine its transmission slot using the device address. Even when the device address is assigned by a centralized sever in the network registration phase, each node autonomously selects its transmission slot, which is the main advantage of TS-LoRa, since no scheduling information needs to be disseminated to the nodes.

Other authors propose a TSCH, which addresses energy efficiency and multi-hop forwarding, over the LoRa approach [15]. Their proposal offers high reliability with collision-free scheduling and channel hopping, but only considers channel or frequency hopping, and does not take advantage of the orthogonality between SFs. The use of a TSCH scheduling and join process results in high protocol overhead for TSCH compared with LoRa.

In a multi-hop or relaying scenario, there must be a scheduling mechanism that provides communication opportunities to the nodes, such that peers can meet at a certain time, frequency, and SF for data exchange. The multi-hop network can be simplified by setting all the nodes to a fixed low SF, as do most of the proposals reported in the literature. However, in harsh propagation environments, some nodes need higher SFs to be able to communicate. On the other hand, higher SFs increase the time on air, duty cycle, and energy consumption. Conversely, when a single SF is used, packet collisions increase, especially for high node densities [16].

The use of TDMA or synchronous schemes for LoRa multi-hop networks has been proposed in the scientific literature [12,17]. A more recent study on a synchronous LoRa mesh [17] focuses on range-critical situations, such as water resource monitoring in urban areas. They present a synchronous LoRa mesh network for the real-time monitoring of critical urban drainage range locations, based on the LoRaWAN MAC protocol with fixed spreading factor and transmission power. The node’s association and synchronization to a particular sub-network is ensured through periodic beacons that are forwarded over several hops of the mesh, following a flooding strategy. Nodes at the same hop level use the same time slot. For scenarios with higher density levels, the number of nodes per level can increase, leading to an increased number of collisions. With a single SF, the adaptive data rate (ADR) feature of LoRa is unused. In [17], precise time synchronization is guaranteed through an optional global positioning system (GPS), or DCF77 long-wave time signaling. This way, neighbors can stay synchronized, allowing a suitable switching between sleeping and waking states within five-minute transmission periods, although this imposes additional constrains for energy consumption.

There are other pioneering multi-hop protocols based on LoRa transceivers, such as LoRaBlink [18], which is low-latency and resilient to interference, but SF hopping is not envisaged. The authors assume a network with low density and low traffic volume, such that their approach does not need a scheduler for organizing transmissions. Their performance study shows a packet delivery ratio (PDR) of 80%.

The research presented in [19] reveals the limits of the LoRaWAN MAC protocol for smart metering applications, showing that due to collisions, PDR is reduced to 25% in networks with very high node densities. To overcome this drawback, a more recent work [20] proposes an energy-efficient network architecture and a highly efficient on-demand time division multiple access (TDMA) communication protocol for IoT. The proposal includes hardware modifications by equipping sensor nodes featuring extra wakeup receiver circuitry with a short communication range. This approach was validated in a test bed experiment using nine nodes that transmitted packets with small payloads (eight bytes) through a mechanism that created a time slot for every node participating in the network. The results showed that both the energy efficiency and the latency of standard LoRa networks were improved, albeit with a more complex and costly design and the implementation of the end-devices.

Table 1 summarizes the advantages and disadvantages of previous works with respect to multi-hop support, the usage of multiple SF, the need for synchronization, and whether the respective authors present an energy consumption analysis. Few of the authors address the energy consumption impact of their proposal. One performs a theoretical energy consumption analysis. Almost all the previous proposals support multi-hop operation, but only some of them include the use of multiple SFs. In this paper, a synchronous mechanism is proposed to solve blind spots based on two-hop LoRa operation using multiple SFs. Our proposal includes an energy consumption analysis that shows the impact of the proposed mechanism on this key parameter. Our proposal does not need a routing mechanism, which reduces the protocol and control message overhead, resulting in energy savings.

## 3. TSSFH Approach for LoRa-Based Networks

This paper introduces a relaying mechanism that complements LoRa-based single-hop networks to mitigate blind spots. The mechanism, called time-slotted spreading factor hopping (TSSFH), combines a TDMA approach with a new channel-hopping method that uses SFs instead of frequencies. The TSSFH mechanism can be combined with both an asynchronous (e.g., classical LoRaWAN) and synchronous (e.g., TS-LoRa [13], TSCH-like [12]) single-hop LoRa-based network, which arranges communication between the GW and the nodes in its coverage area.

Our approach considers nodes that can reach the gateway and do not relay any messages for other nodes. These are called connected nodes (CNs) and behave similarly to normal LoRa nodes, but with an adaptation that allows them to change their role to relay nodes (RNs). We also consider disconnected nodes (DNs) in blind spots that need to find a relay node to reach the base station. If DNs do not find a relay node, we call them isolated nodes (INs), as shown in Figure 1.

With our TSSFH proposal, a node in a blind spot (i.e., outside the GW coverage area due to distance or channel impairments) can use multiple SFs to communicate with a node in a GW range, with the latter acting as a relay between the disconnected nodes and the GWs. The main purpose is to deliver a packet to the relaying node (RN) using the proposed TSSFH MAC protocol. This results in a two-hop LoRa-based network, as shown in Figure 1. The advantages of using multiple SFs are twofold: (i) when a node is disconnected within a blind spot, it may reach a relaying node using a lower SF, but it may need a higher one; and (ii) the use of multiple SFs increases the range of possibilities and the space for communication opportunities.

Figure 2 shows a flowchart illustrating the different roles that nodes can play in a LoRa-based network that uses TSSFH to mitigate blind spots. When a node in the network sends a message to the GW, this node acts in a very similar way to a classical LoRaWAN node. The difference lies in the fact that it extends its reception windows to increase the probability of receiving beacon requests from nodes inside the blind spots. During the association process, nodes inside blind spots listen for transmissions from CNs or RNs to send them back beacon requests (after detecting transmissions from them). This occurs during the extension period, just when a CN (or a RN) finishes sending a message and after receiving its respective acknowledgement from a GW. Once a beacon request is received by a CN, it switches its role to that of a relay node activating TSSFH, which periodically opens extra listening windows for relaying purposes. RNs relay data packets from nodes inside blind spots. Every time a packet is received, the RN timer is reset, extending the time that the node stays in the role of RN. After a certain time without receiving any packets from associated nodes (RN timer timeout), the RN returns to its previous and basic role of CN.

If a node fails to reach a gateway (i.e., if no ACK is received), it assumes that there are no gateways in its range. These nodes, which are called disconnected nodes (DN), start the TSSFH association process, which is described by Algorithm 1. The association process is based on a passive scanning strategy, in which each node listens for incoming packets from any CN or RN during the *N_p_* transmission periods (*T_btwTx_*) on each SF. *N_p_* and *T_btwTx_* are design parameters. The parameter *N_p_* depends on the time needed by DNs to associate and the transmission period depends on the frequency at which the application needs to send sensed data. If a DN detects a data packet sent by a RN or CN, it sends an association beacon request during the parent node (CN or RN)’s extended listening window. The parent node is added to the parent list in case a successful association beacon is received as a response. If, during the passive scanning strategy, a DN detects any association beacon response, the RN sender of this response message is directly added to the DN’s list of parents (list of associated RNs). This is also the case if the association beacon request is sent by another disconnected node requesting association.
**Algorithm 1** TSSFH Association**Inputs**: *T_btwTx_*, *N_p_*1 set radio mode: Rx2 **for each** SF∈[SFmin,SFmax]3  set TSF timer to NP∗TbtwTx4  **if** detects packet from RN or CN **then**5     send association beacon request6  **end if**7  **if** receives association beacon **then**8     add RN of CN to parent list9  **end if**10   **if** TSF timer expired **then**11      go to next for each iteration12   **end if**
13 **end for each**

Once the association timer expires, each node checks its parent list, and when at least one parent is available, the node takes the role of DN. The DNs send data packets using the TSSFH communication mechanism. Nodes that cannot find a parent during the TSSFH association process take the role of isolated node (IN) and turn off for a given period before trying to obtain access to the network again.

The RNs, also called listening nodes, can be power-constrained devices and listen during specific time slots. TDMA structures these time slots and makes it easier to estimate when relaying nodes will listen. As it is possible to have many relaying nodes, several of them can listen on the same SF, but using different time slots. Two random selection mechanisms help to avoid collisions: (i) the relaying nodes randomly select a cell (tuple of frequency, SF, and time slot) to listening to the complete set of cells available; and (ii) nodes in a blind spot randomly select a cell in this subset, where relaying nodes listen.

To design the proposed TSSFH, let us assume that the transmission of a packet, using SF7, has a time on air (ToA) of T seconds. Considering LoRa’s ADR, this means that using SF8, it will take about 2T seconds, about 4T for SF9, and so on, up to SF12, which takes about 32T seconds. Ensuring T is big enough to guarantee any LoRaWAN packet transmission, we can define a frame structure with a duration of 32T seconds. This means that nodes hopping to different SFs can find a different time-slot structure, as shown in Figure 3.

The proposal in Figure 3 is only an ideal approach, and it does not consider the maximum payload allowed for every SF in the LoRaWAN MAC protocol, to ensure back-compatibility with legacy LoRa-based networks. The maximum payload size is different according to the SF: 51 bytes for SF10, SF11, and SF12; 115 bytes for SF9; and 242 bytes for SF8 and SF7 [21]. Figure 4 shows the proposed time slot.

A guard time of 30 ms is assumed (some authors have considered 15 ms [13] and 22 ms [14]). This implies that considering 50 ppm as a maximum clock drift [22], a synchronization period of 10 min is a good estimate. Time intervals of 19.7 ms [14] to switch the radio nodes from transmission to reception in the DNs, and the opposite in the RN, are common values. The ACK packet and the synchronization beacon have a payload of three bytes, which include two bytes for synchronization and one byte to indicate the message type, as shown in Figure 5. The type field depends on the TSSFH message type: 0 (association beacon request), 1 (association beacon response), 2 (ACK), and 3 (synchronization beacon). As is explained further later in this section, the cell index (*Icell*) and the frame’s number (*No_frame_*) are shared by the RN with its child DN in the association beacon’s response. The association beacon request message only needs one byte of payload to specify the message type.

Taking the computed ToA for the maximum payload size [23] and the proposed slot structure, the slot duration for each SF is obtained as shown in Table 2. As some SFs have different maximum payloads, their ToAs can be similar. Extra guard time is added and timeslot durations can again be related by powers of two (1, 2, 4 and 8). As such, we can propose four different slot durations (0.6, 1.2, 2.4 and 4.8 s).

Using these particularities, a more practical approach can be proposed with a frame structure of 23 cells (tuple of Frequency, SF and time slot) lasting 4.8 s, as shown in Figure 6.

All the RNs use the same vector of cells, *V_cell_*, which is shared among the whole network. The vector is stored in the node’s firmware during configuration. The following example shows the full vector of 23 cells when the six SFs are used: *V_cell_* = {0, 8, 12, 1, 16, 20, 2, 9, 13, 3, 17, 22, 4, 10, 14, 5, 18, 21, 6, 11, 15, 7, 19}. Considering that higher SFs increase the time on air, duty cycle, and energy consumption, and that SF11 and SF12 only contribute three cells, cells 20, 21, and 22 were removed from the vector list.

The RNs randomly select a cell index (*I_cell_*) that is shared with their DN neighbors during the association process. This allows nodes in blind spots to predict the cells on which their potential relaying nodes will listen during their listening windows, because the nodes scan the vector following a round-robin strategy. Listening windows implement a multi-frame structure, in which every frame can have a maximum of 20 cells (using SF7–SF10), according to the number of SFs in use. As a result, multiple RNs can listen at the same time, albeit using a different SF, or on the same SF, but at different time slots. Figure 7 shows a sequence of three listening windows implementing just one frame (20 cells each) and the cells scanned by three RNs.

The transmission period and the number of listening windows to be opened are important parameters in the TSSFH approach. The transmission period, also called the notification period, is the average time interval between two consecutive data transmissions. The transmission period depends on the frequency needed by applications to send measurements to the application server. On the other hand, the number of listening times between transmissions, which represents the number of listening windows to be opened by the RNs in every transmission period, influences the network performance and energy consumption, as discussed in the next section. These design parameters make it possible to compute the listening window period (*T_btwLn_*) by using Equation (1):(1)$$T_{btwLn}=\frac{T_{btwTx}}{N_{LnPerTx}},$$
where *T_btwTx_* is the transmission period of the application and *N_LnPerTx_* is the number of listening windows opened in every transmission period.

Each potential DN can transmit in any of the *N_LnPerTx_* listening windows opened, during which potential parent RNs listen to relay packets towards a GW. The listening window period is used to predict the cell time (*T_Cell_*) in which the RNs will listen, as shown in Equation (2):(2)$$T_{Cell}=AWN*T_{btwLn}+4.8*No_{frame}+Cell_{offset},$$
where *AWN* is the absolute window number, a counter that stores the number of windows that have occurred since the network start-up; *No_frame_* is the frame number in the multi-frame structure, and *Cell_offset_* is the time offset of the cell within the frame structure, as shown in Figure 6. Figure 8 depicts an example of how *T_Cell_* is determined for two different cells, cell A and cell B, with *T_Cell_* called tA and tB, respectively. The figure represents two transmission periods, in which a multi-frame structure of three frames is used. 

## 4. Energy Consumption Analysis

In this section, we discuss the energy impact of TSSFH. We do this by calculating the average current draw in a transmission period. This average current draw is proportional to the average power consumption in a transmission period (*P_ave_ = I_ave_ × U*, as *U* is considered constant). We consider a periodic situation in which the same actions are repeated in every transmission period; therefore, these averages per period stay the same over time. To calculate the energy consumption during a certain time, it is necessary to multiply the average power consumption by the elapsed time. The elapsed time is an integer multiple of the transmission period, since time is a sequence of transmission periods.

Our study considers the average current draw based on the model developed in [24]. The RNs open multiple listening windows in every transmission period to receive and relay packets from the DNs. The more listening windows are open, the better the expected performance in terms of PDR, as the DNs can have more transmission opportunities when enough relaying nodes are available. However, a higher number of listening windows implies more current draw. Therefore, the optimal number of listening windows is estimated and the average current draw for the different listening rates is evaluated.

The “states” of a node involved in the TSSFH communication process differ from those traversed in other LoRa-based MAC protocols. It is assumed that the LoRa sensor nodes will display periodic behavior; therefore, the current draw is modeled during one transmission period. The MultiConnect mDot platform from Multitech [25], based on the SX1272 transceiver [22], was used for the models in [24], measuring the magnitude of the current drawn in each “state”. Table 3 describes the different states traversed by nodes during a communication process and includes values for state duration and current draw.

According to Table 3, the average current draw in a period between transmissions can be computed by Equation (3):(3)$$I_{ave}=\frac{1}{T_{btwTx}}\overset{S}{\underset{k=1}{\sum }}T_{k}*I_{k},$$
where *S* is the set of states in Table 3, with a corresponding duration and current draw *T_k_* and *K_i_*, respectively. The sleep time (10th state) is calculated by Equation (4):(4)$$T_{sleep}=T_{btwTx}-T_{act},$$
where *T_act_* represents the activity time as the sum of the rest of the states during the communication process.

The duration of the transmission and reception times *T_tx_*, *T_rx_*, and *T_rx_idle_*, depends on other factors, such as the payload and data rate. The reception time when no preamble is detected (idle listening), can be determined by Equation (5):(5)$$T_{rx_{idle}}=N_{dsym}*T_{sym},$$
where *T_sym_* is the time on air to transmit a LoRa symbol and *N_dsym_* is the number of LoRa symbols the end-device keeps receiving while waiting for preamble detection.

Table 4 summarizes the values of *T_tx_*, *T_rx_*, and *T_rx_idle_*, assuming eight symbols as the preamble length, a coding rate (CR) of 4/5 (except for the 20 bits of the physical header, for which a CR of 4/8 is used), and a bandwidth (BW) of 125 kHz [24]. The *T_tx_* is obtained by considering two scenarios with payloads of 50 and 100 bytes [27], respectively.

The TSSFH has a sequence of states that depends on the process to be followed by the sensor nodes, according to the roles they play. The DNs transmit a packet to and receive an ACK from the RNs; the RNs receive a packet from and transmit an ACK to the DNs or perform idle listening. The CNs extend the reception window opened after every transmission, allowing unassociated nodes to join the network. The sequence of states involved in the TSSFH approach can be managed more easily as an independent process, as follows:

DNs:

Sleep → Wake up → Radio preparation → Transmit packet → Radio switch → Receive ACK → Radio off → Postprocessing → Turn-off sequence → Sleep.

RNs:

Sleep → Wake up → Radio preparation → Guard time → Receive packet → Radio switch → Transmit ACK → Radio off → Postprocessing → Turn-off sequence → Sleep.

RN*_idle_*:

Sleep → Wake up → Radio preparation → Guard time → Idle listening (*Trx_idle*) → Radio off → Postprocessing → Turn-off sequence → Sleep.

In our analysis, we consider two applications with payloads of 50 and 100 bytes, which are able to use the spreading factors SF7–SF10 and SF7–SF9, respectively. The values of *T_tx_*, *T_rx_*, and *T_rx_idle_* for the applications are summarized in Table 4. Pseudo-random scheduling is assumed. The average packet transmission and reception times can be computed by Equation (6):(6)$$T_{tx}=T_{rx}=\frac{1}{CellO}\underset{\forall i}{\sum }NS_{SF_{i}}*ToA_{SF_{i}}$$
where *ToA_SFi_* is the time on air of *SF_i_*, as shown in Table 4; *CellO* is the number of cells in the frame (*CellO* = 20 for a 50-byte application (SF7–SF10) and *CellO* = 16 for a 100-byte application (SF7–SF9), according to Figure 6; *i* is the SF index (i.e., *i* = 7, 8, 9, 10) and *NS_SFi_* is the number of slots for the *SFi*. According to Figure 6, *NS_SFi_* can be determined as follows:(7)$$NS_{SF_{i}}=\{\begin{array}{c} 8\\ 4 \end{array}\;\;\;\begin{array}{c} i=7\\ \;\;\;\;\;\;\;\;i=8,9,10 \end{array}$$

Substituting Equation (7) and the values from Table 4 into Equation (6), we have *T_tx_Ave_* = 308.2 ms for a 50-byte payload and *T_tx_Ave_* = 333.2 ms for a 100-byte payload. Considering these values and the 1% duty cycle restrictions, the minimum time between transmissions can be computed as 99**T_tx_Ave_*_,_ which means that 31 s and 33.3 s are the minimal times between transmissions for the 50-byte and 100-byte payload sizes, respectively. Even when the prior analysis does not consider the ACK packet or synchronization packet transmissions, it reveals useful insights into the transmission period’s lower bound. Note that the ACK and synchronization packet sizes are much smaller than the data packet sizes.

As mentioned above, the LoRa-based nodes using TSSFH can perform three communication processes: DN packet transmission, RN packet reception, and RN idle listening. We can compute the total duration for each communication process, as shown by Equations (8)–(10):(8)$$T_{DN}=T_{wu}+T_{pre}+T_{tx}+T_{switch}+T_{rx_{ACK}}+T_{off}+T_{post}+T_{seq},$$
(9)$$T_{RN}=T_{wu}+T_{pre}+T_{guard}+T_{rx}+T_{switch}+T_{tx_{ACK}}+T_{off}+T_{post}+T_{seq},$$
(10)$$T_{RN_{idle}}=T_{wu}+T_{pre}+T_{guard}+T_{rx_{idle}}+T_{off}+T_{post}+T_{seq}.$$

Assuming the case in which the network can have as many RNs available as the number of detached DNs, the RNs receive one packet in every transmission period and perform idle listening in the rest of the listening windows. The sleeping time between transmissions (*T_sleep_*) is then obtained as the result of combining Equations (4) and (8)–(10), as follows:(11)$$T_{sleep}=\{\begin{array}{c} T_{btwTx}-T_{DN}\\ T_{btwTx}-N_{RxPack}*T_{RN}-\left(N_{LnPerTx}-N_{RxPack}\right)*T_{RN_{idle}} \end{array}\begin{array}{c} \;\;\;\;\;\;\;\;DN\\ \;\;\;\;\;\;\;RN \end{array}$$
where *N_RxPack_* is the number of packets received by the RNs. By combining Equation (3), Equation (11), and the results in Table 3 and Table 4, the average current draw can be computed.

Based on the analysis described above for the TSSFH approach, we can estimate the average current draw per transmission period, in which the DNs transmit one message and the RNs potentially receive one message. For the RNs, it is assumed that in only one of the listening windows, a message is received, while the remaining windows perform idle listening. The RNs also send an extra beacon (equivalent to an ACK packet) when the transmission period is greater than 10 min, the synchronization period that we assumed, as the ACK packets also carry synchronization information.

As mentioned above, the CNs extend the reception window after a LoRa transmission to offer association opportunities to the DNs. The window duration can be adjusted to reduce energy consumption at the expense of offering fewer association opportunities to the detached nodes or new ones out of the GW range that want to join the network.

Figure 9 shows different curves corresponding to the average current draw for several listening frequencies, considering an application with 100 bytes of payload length. The behavior is shown for a range between 1 and 10 listening windows per transmission period, the latter being represented in a range between 1 and 1000 min. To facilitate the comparison, we take as benchmark the curve in the dashed line that depicts the LoRaWAN single-hop average current draw for SF7, considering acknowledged transmission [24].

It can be seen that the curves that open five and six listening windows are comparable with the benchmark used. Similar results representing the current analysis for a 50-byte payload application, can be observed in Figure 10.

## 5. Theoretical Performance

There are several metrics that can be used to measure the performance of the proposed LoRa-based approach for nodes inside blind spots. Nodes inside blind spots use neighboring relaying or listening nodes (RNs) to reach a gateway. The greater the number of RNs available in a network segment and the greater the number of listening windows that RNs can open in every transmission period, the greater the packet delivery probability of the transmitter nodes (DNs), since more communication opportunities are available to them, thus improving the performance of the LoRa-based network. Based on the energy analysis carried out in the previous section, it was estimated that the RNs participating in the proposed LoRa-based network can open a maximum number of six listening windows per transmission period (*N_LnPerTx_* = 6), in order to maintain an energy consumption comparable with the single-hop LoRaWAN. This way, each RN listens six times per transmission period, using any of the available cells (frequency, SF, TS) known for the transmitter nodes in the blind spot, once they are associated into the network by receiving a beacon frame from their parent relaying neighbors (RNs). The beacons should include, among others, information related to network synchronization and the RNs’ cell indexes.

To better illustrate the nodal activity, Figure 11 shows the interaction of nine nodes (four relaying nodes and five nodes in a blind spot) during two transmission periods in a network operating with three listening windows per transmission period (*N_LnPerTx_* = 3), each with a single frame. One of the nodes in a blind spot, DN1, achieved a successful transmission in the first transmission period, as no other node was using cell 7. However, in the second transmission period, a collision occurred, as DN3 was also using cell 5. The relaying nodes RN1 and RN4 were overhearing, as both were using the same cells for listening, duplicating the reception of messages transmitted by DN2 in the second listening window. The collision originated by DN4 and DN5 during their transmissions in the second transmission period also caused overhearing, as RN1 and RN4 were listening in the same cell 8 (making duplicates of either transmission or collision). As expected, there was a significant amount of idle listening, as each relaying node listened three times in every transmission period, but some (RN1 and RN4) were duplicated, potentially causing overhearing, a situation that was repeated in every listening window.

During association, an RN randomly selects one cell to listen, from 20 ∗ *frames* different cells (considering the maximum number of 20 cells per frame). Once the first selection is made, the RN uses its cell index and the vector of the cells to select the next cell for listening in the next listening windows. As different RNs can select the same cell to listen, the *L_RN_* listening nodes have ${\left(20*frames\right)}^{L_{RN}}$ ways to select their listening cell. Meanwhile, the ways to choose a different cell can be calculated with Equation (12):(12)$$\frac{\left(20*frames\right)!}{\left(20*frames-L_{RN}\right)!}.$$

Consequently, the probability that each listening node selects a different cell is:(13)$$P_{DiffCell}=\frac{\left(20*frames\right)!}{{\left(20*frames\right)}^{L_{RN}}\left(20*frames-L_{RN}\right)!},$$
where (X)! is the factorial of X.

From Equation (13), *P_DiffCell_* decreases when the number of relaying nodes (*L_RN_*) increases. However, even when *L_RN_* is large, some RNs can select the same cell for listening, reducing the relaying possibility of the presence of nodes in blind spots. Let us define *L_cell_* ≤ *L_RN_* as the number of different listening cells selected by *L_RN_* relaying nodes. Considering that in a transmission period *N_LnPerTx_* listening windows are opened, each transmitter node, DN, has *L_cell_* ∗ *N_LnPerTx_* different listening cells to transmit. Using a similar reasoning to Equation (13), the probability that *L_DN_* transmitter nodes in a blind spot do not collide (i.e., select a different listening cell) is:(14)$$P_{NoColl}=\frac{\left(L_{cell}*N_{LnPerTx}\right)}{{\left(L_{cell}*N_{LnPerTx}\right)}^{L_{DN}}\left(L_{cell}*N_{LnPerTx}-L_{DN}\right)!}.$$

However, for any of the nodes in a blind spot, the successful transmission probability can be computed as:(15)$$P_{Succ\_1Cell}=\frac{1}{L_{cell}*N_{LnPerTx}}{\left(\frac{L_{cell}*N_{LnPerTx}-1}{L_{cell}*N_{LnPerTx}}\right)}^{L_{DN}-1},$$
where the first term in Equation (15) is the probability that any node in a blind spot selects one of the available listening cells, and the second term is the probability that the other *L_DN_* − 1 transmitters select any of the remaining *L_cell_* ∗ *N_LnPerTx_* − 1 listening cells. The collision-free packet delivery ratio (PDR) for any cell (i.e., successful transmission in any cell) can be computed as:(16)$$\mathrm{PDR}=P_{Succ\_1Cell}*\left(L_{cell}*N_{LnPerTx}\right)={\left(\frac{L_{cell}*N_{LnPerTx}-1}{L_{cell}*N_{LnPerTx}}\right)}^{L_{DN}-1}.$$

Figure 12 illustrates the performance of the theoretical PDR with *N_LnPerTx_* = 6, under different conditions of *L_DN_* and *L_cell_*. Just as expected, the PDR increased with the number of available cells and decreased with the increment of the transmitters (DNs).

We assume that there is no coordination between the RNs, and that each RN individually defines the time to open its listening window, as Figure 13a shows. Asynchronous behavior (i.e., a lack of coordination) between the RNs can cause a typical collision between nodes that are transmitting to different RNs in a blind spot, as shown in Figure 13b. However, the worst case happens when the listening windows of different RNs are perfectly aligned (Figure 13c); in such cases, collisions can occur in any cell of the listening window.

Both the analytical and MATLAB simulation performance measures assessed in this paper addressed the worst case shown in Figure 13c. The analytical results of the PDR shown in Equation (18) did not consider the collision probability generated by the RN or CN transmissions in the single-hop network (e.g., LoRaWAN). Estimating or simulating collisions between LoRaWAN (or another MAC protocol used in the single hop network) and TSSFH is more complex, so it is also addressed in the next section, using OMNeT++ simulation scenarios.

## 6. Performance Evaluation

The performance evaluation of the proposed TSSFH mechanism was carried out in two different scenarios: (i) isolated blind spot scenario and (ii) LoRA network with multiple blind spots. The first scenario was used to evaluate the performance of the proposed TSSFH considering neither the influence from other blind spots nor that of the CNs in the network, and the PDR was evaluated using different combinations of DNs and RNs. The LoRa network with multiple blind spot scenarios was used to evaluate the performance of the TSSFH in realistic LoRa networks considering the influence of LoRaWAN traffic and multiple blind spots.

### 6.1. Isolated Blind Spot Scenario

The MATLAB programming language tool was used to develop Monte Carlo experiments. The experiments considered a single blind spot with multiple DNs served by multiple RNs. In each experiment, a number of listening windows was opened by each RN, which uniformly selected a cell to listen for a DN transmission.

In every transmission period, the transmission nodes (DNs) also used a discrete uniform distribution function to randomly select one of the available listening cells for sending their data packet. The simulation time for each scenario is 8 days. Using a transmission period of 15 min this results in 768 transmission periods. The results in the following sections are the average of 500 independent runs.

The main metric proposed to determine the network performance was the average PDR, considering only the impact caused by the presence of collisions during the transmission periods. Other kinds of signal perturbation caused by the presence of noise, attenuation or interference, are not covered in this research. Metrics such as the average time spent listening idly and overhearing were also obtained in order to determine the utilization of communication resources and their impact on power consumption. The PDR was computed as:(17)$$\mathrm{PDR}=\frac{\sum^{\;}packet_{collisions}}{L_{DN}*Simulation_{Periods}}.$$

Equation (17) also considered the influence of collisions caused by LoRaWAN transmissions of RNs belonging to the blind spot, because they could have interfered with the TSSFH transmissions of the DNs inside the blind spot. The packet collisions were calculated as the sum of the DNs that selected the same cell in a transmission period plus any TSSFH packet that collided with any LoRaWAN transmission from the RNs. *Simulation_Periods_* denotes the number of transmission periods during the total simulation time.

The number of overhearing nodes was computed as the number of nodes that received the same packet (in the same cell), resulting in extra energy consumption in the RNs, since two or more of them received the same packet. Therefore, Equation (18) computes the average overhearing per node:(18)$$Overhearings\_Per\_Node=\frac{\sum^{\;}duplicated_{packets}}{L_{RN}*Simulation_{Periods}}$$

Finally, the average idle listening per node was the sum of the listening cells opened by the RNs that were not selected by any of the transmitter nodes (DNs) per transmission period, as shown in Equation (19):(19)$$Idle\_Listenings\_Per\_Node=\frac{\sum^{\;}idle_{listenings}}{L_{RN}*Simulation_{Periods}}$$

It was expected that increasing the number of frames would result in a greater diversity in the random selection of cells that were chosen by the RNs when opening listening windows, reducing the probability of coincidences and corresponding overhearing. In theory, using an infinite number of frames, the highest possible PDR could be obtained according to the number of RNs available (*L_RN_*) to relay traffic from the transmitter nodes (*L_DN_*) disconnected inside blind spots. However we expected that after a certain number of frames, the improvement in the PDR would become negligible. On the other hand, the larger the number of frames in each listening window, the longer the lower limit for the transmission period, decreasing the frequency with which the applications could send data. The selection of the number of frames would be a trade-off between the quality of service in terms of PDR and the minimum data transmission frequency of the applications. Therefore, our first target was to determine a key value for the number of frames from which no appreciable or significant improvement in PDR would be observed.

To determine the impact of the number of frames (multi-frame structure implemented by TSSFH) on the PDR performance, a first simulation was carried out for different combinations of *L_DN_* and *L_RN_*, so that a sufficient number of different combinations could be simulated, representing a wide range of cases whose results could be generalized. For a range from 1 to 64 frames, each scenario computed the average PDR. Figure 14 shows the PDR in terms of the number of frames for each of the twelve selected scenarios for a range between 1 and 25 frames, where more representative improvements in PDR were to be found. The simulations considered that the RNs also transmitted their data in the LoRa network. The theoretical results given by Equation (16) are also shown; the parameter *L_cell_* in Equation (16) was computed by simulation.

The analytic results from Equation (16) and the simulation results matched well. Based on the analytic model developed in Section 5, the influence of the RNs belonging to the blind spot on the PDR could be estimated, resulting in a maximum difference of 0.86% when one frame was used. The points at which the PDR improvement began to dip beneath 0.05% are highlighted with a star on each curve. Eleven frames were used in the rest of the experiments; however, using more frames did not result in a significant improvement.

The second MATLAB simulation aimed to determine the number (*L_RN_*) of parent nodes (RNs) needed to relay the traffic of the disconnected nodes in a blind spot (*L_DN_* transmitter nodes), for three cases in which the PDR remained equal to or greater than 90%, 95%, and 98%, the latter being a typical PDR value for smart metering applications in smart-grid power systems [25]. To represent several of the possible scenarios, the experiment went through them one by one, increasing the number of transmitters until a maximum of 35 transmitter nodes was reached. It was assumed that *L_RN_* would never exceed 64 RNs.

The experiment employed a vector of cells with twenty indexes (*I_cell_*), as illustrated in Figure 7 in Section 3, six listening windows, and a multi-frame structure of eleven frames. We assumed that a collision would occur when two or more transmissions coincided in frequency, time, and SF, meaning that they would select the same transmission opportunity. Figure 15a shows the results of the experiment, determining the number of listening nodes needed to relay the traffic generated by the maximum number of possible transmitter nodes L_DN_ guaranteeing a PDR equal or greater than 90%, 95%, and 98%. For each of these pairs (*L_RN_* vs. *L_DN_*), the average overhearing and idle listening values (per node) per transmission period were obtained, as Figure 15b,c show.

The results shown in Figure 15a reflect exponential behavior, which is typical for random access. To achieve a higher PDR, we expected to observe a high proportion between the relaying nodes and the disconnected nodes. For instance, if we compare the listening nodes needed to relay the traffic generated by seven transmitter nodes in a blind spot, we need 10, 21, and 58 listening nodes for a PDR of 90%, 95%, and 98%, respectively. This means that to increase the PDR for any application, the number of required listening nodes needs to increase exponentially. As expected, Figure 15b shows how the overhearing increased almost linearly with the number of transmitter nodes.

The idle listening showed similar behavior in each of the cases considered in the experiment. Figure 15c shows the average idle listening per node in terms of the transmission period, resulting in values of 5.38, 5.68, and 5.86 for a PDR of 90%, 95%, and 98%, respectively. These results corresponded with the increment of RNs needed to obtain the target PDR. The increment in the number of RNs resulted in a reduction in the number of average retransmissions per RN, meaning that the average idle listening tended to approach its upper value. Given that the nodes opened six listening windows in every transmission period, the average idle listening values had an upper limit equal to the number of listening windows opened.

### 6.2. LoRA Network with Multiple Blind Spots

A simulation environment was developed using the OMNeT++ simulator to evaluate the operation of the TSSFH mechanism under more realistic conditions. The simulation model is based in the Flora framework [26], which provides an implementation of LoRaWAN and the modules that simulate the physical layer of LoRa. The module that simulates the operation of the TSSFH mechanism was created, and it was used as a complement to the Flora library. The main objective of the experiments was to determine the PDR of the nodes communicating with the TSSFH and the influence of the collisions with LoRaWAN and other blind spot transmissions. For a node in the blind spot that uses the proposed TSSFH, there is no difference between the interference generated by a node belonging to the home LoRa network, but located outside the blind spot, and the interference caused by a LoRa neighboring network, or even by non-LoRa networks that use the same spectral band. We considered a scenario with one gateway and 150 nodes. The simulation time was limited to 8 days. The end devices were randomly distributed around the gateway in a 7000 m × 7000 m area, and they used the same vector of cells as defined in the previous sections. Table 5 shows the parameters used in the simulation.

Figure 16 shows the simulation scenario. In addition to one-hundred-and-fifty nodes, three blind spots were manually placed, with three, six, and nine nodes, respectively, with the parameters listed in Table 5. According to the simulation experiments, these blind spots were attended by 11, 25, and 35 RNs, respectively.

The PDR for each blind spot was determined and compared with the expected theoretical value obtained from Equation (16). The results are shown in Table 6. The theoretical model does not consider the LoRaWAN influence, which was captured by the simulation, causing the simulated results to be slightly worse than the theoretical results.

To determine the influence of the LoRaWAN traffic over the TSSFH communication, the PDR was evaluated for two different simulation cases. One considered only the TSSFH traffic (turning off the LoRaWAN traffic outside the blind spots), and the other considered both the TSSFH and all the LoRaWAN traffic. As explained in previous sections, our proposal targets blind spots, so the PDR is a measure of the packets successfully delivered to the RNs, since TSSFH is used to complement the LoRa-based network (e.g., LoRaWAN, TSCH-like, TS-LoRa, etc.). The evaluation of the PDR in gateways would be highly influenced by the network protocol (LoRaWAN in this case) and would not properly illustrate the TSSFH’s performance. The experiment was carried out by varying the number of nodes in the network between 150 and 650, but maintaining the same number of DNs shown in Figure 16 to analyze the performance. This will allow to evaluate how the improvement in PDR, obtained by increasing the number of RNs that attend a blind spot, is affected by increased network density, entailing aggravated LoRaWAN influence. The number of RNs for each blind spot is shown in Figure 17. As expected, the number of RNs increased along with the network density, improving the communication opportunities for the nodes isolated within the blind spots.

Figure 18 shows the PDR at the blind spot with nine DNs, which is the one with worst performances across the three blind spots, with and without the influence of LoRaWAN transmissions and the expected PDR from the theoretical model.

The results show a match between the theoretical model and the simulations without the LoRaWAN’s influence. The theoretical model had a slightly better performance than the simulations because the theoretical model does not take into account the interference between the different blind spots. The TSSFH packet delivery ratio was mainly determined by the number of nodes within the blind spots, and improved slightly with an increasing number of RNs, as a result of the increase in the network density. The influence of LoRaWAN traffic became significant as the node density increased in the simulated network; a PDR value close to 75% was obtained when the network reached a total of 650 nodes. The rest of the blind spots demonstrated similar behavior, as shown in Table 7.

## 7. Conclusions and Future Work

This paper proposes a new medium-access control (MAC) mechanism as a complement to the MAC protocols used in LoRa-based single-hop WANs, to mitigate the presence of blind spots. Unlike other multi-hop based mechanisms, TSSFH takes advantage of the orthogonality present in LoRa virtual channels by using all the available spreading factors. This makes it possible to offer greater communication opportunities for devices that are outside the range of a GW. TSSFH uses a relaying mechanism that allows it to mitigate the blind spots in any LoRa-based single-hop networks that use asynchronous (e.g., LoRaWAN) or synchronous (e.g., TS-LoRa, TSCH-like) MAC protocols, complementing its operation. The use of a relaying mechanism converts a single-hop LoRa-based network into a two-hop network for nodes outside the GW’s range, which considerably reduces the protocol overhead as no routing protocol is needed. This type of reduction is indispensable in general multi-hop networks. Unlike multi-hop approaches, which use a single SF, TSSFH can use all available SFs, which increases the range of possibilities and the space for communication opportunities.

The TSSFH uses a pseudo-random schedule for the selection of communication opportunities, which simplifies implementation and network organization. An increase in PDR assumes a proportional increase in the number of listening nodes present in the network to perform the relaying function. The opening of listening windows by the relaying nodes offers communication opportunities for the nodes in the blind spots. The more listening windows the RNs open, the more communication opportunities the nodes in the blind spot have. Therefore, the nodes deliver packets more effectively, albeit with higher energy consumption in the RNs. The energy analysis we carried out points to a balance between nodes’ delivery performance in the TSSFH domain and their energy consumption, if the energy consumption of LoRaWAN nodes are taken as a benchmark. The results show that the power consumption for the nodes within the blind spots was comparable to the LoRaWAN nodes’ consumption when six listening windows were opened. In the case of nodes acting as relaying nodes, this consumption increased proportionally to the number of messages they were required to forward. This affected the lifetime of these nodes (RNs), but it was necessary in order to avoid the installation of new GWs. If such nodes can be well identified in fixed locations, it is recommended to use energy scavenging for them.

With the increase in the network density, the number of relaying nodes also increased, which entailed an increase in the transmission opportunities and, consequently, an increase in the PDR. The PDR remained above 95% in the three blind spots simulated in the scenario with 150 LoRaWAN nodes with a deterministic traffic characterized by a payload of 50 bytes and a transmission period of 15 min. However, when the density of the nodes increased significantly, the influence of the LoRaWAN traffic began to affect the performance of the network, causing the PDR to drop below 90% when the number of nodes exceeded 450, that is, when the network density was increased threefold. Therefore, we can conclude that the use of TSSFH in very-high-density LoRa-based WANs might reach moderate PDR values for nodes that are completely disconnected from the network in blind spots.

Despite this paper, we are still far from concluding our research. Future work will aim to extend TSSFH to full multi-hop implementation, including time and routing protocols considering a more efficient use of SFs. Further developments of TSSFH must include its evaluation in more practical and real-world scenarios, addressing perturbations caused by noise, attenuation, and interference.

## Figures and Tables

**Figure 1 sensors-22-02253-f001:**
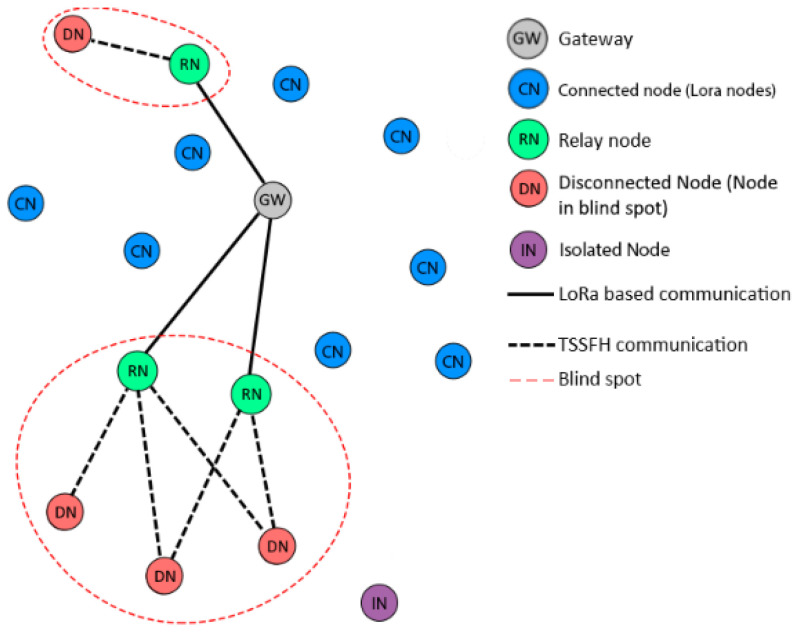
TSSFH complementing a LoRa-based network to mitigate blind spots.

**Figure 2 sensors-22-02253-f002:**
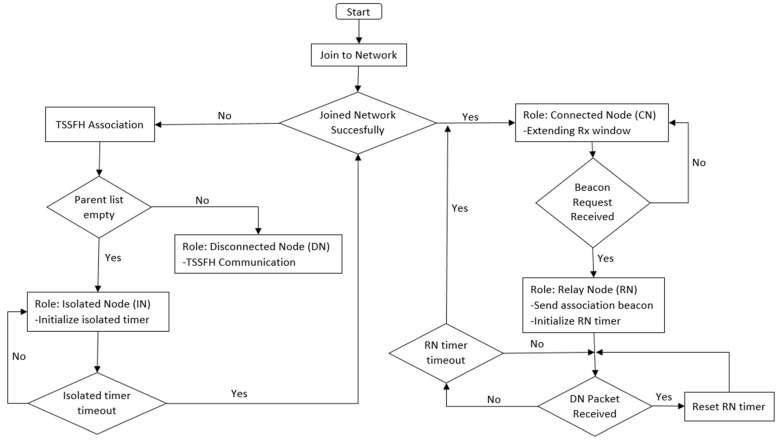
Flowchart showing the role changes of nodes.

**Figure 3 sensors-22-02253-f003:**
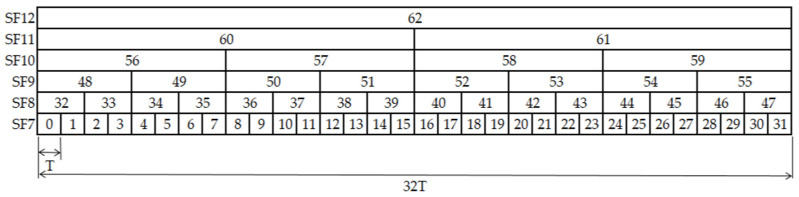
Ideal frame structure in TSSFH approach.

**Figure 4 sensors-22-02253-f004:**
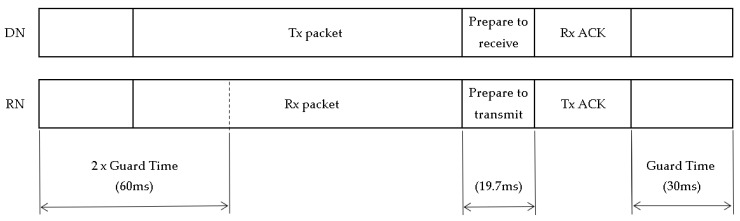
Time-slot structure.

**Figure 5 sensors-22-02253-f005:**
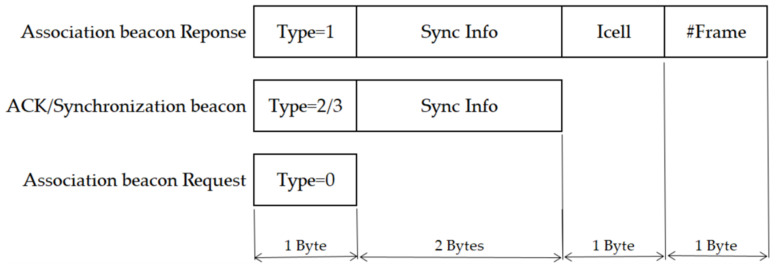
Beacons’ payload format.

**Figure 6 sensors-22-02253-f006:**
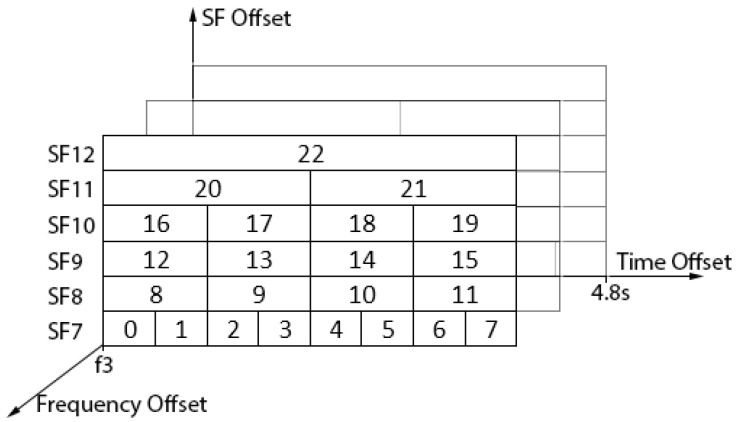
Frame structure in TSSFH.

**Figure 7 sensors-22-02253-f007:**
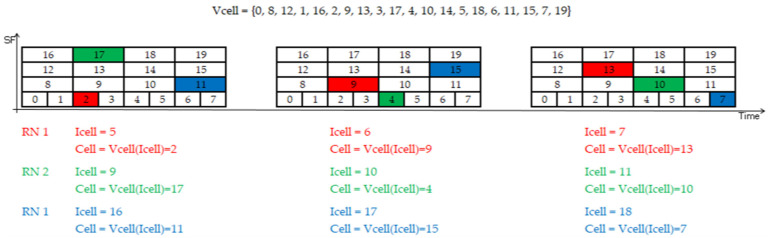
Cells scanned by three relay nodes during three consecutive listening windows.

**Figure 8 sensors-22-02253-f008:**
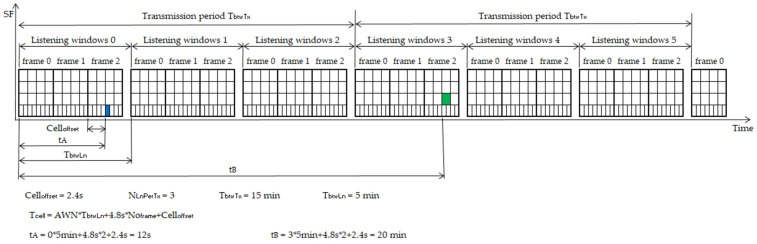
*T_cell_* calculated for cell A and B being tA and tB.

**Figure 9 sensors-22-02253-f009:**
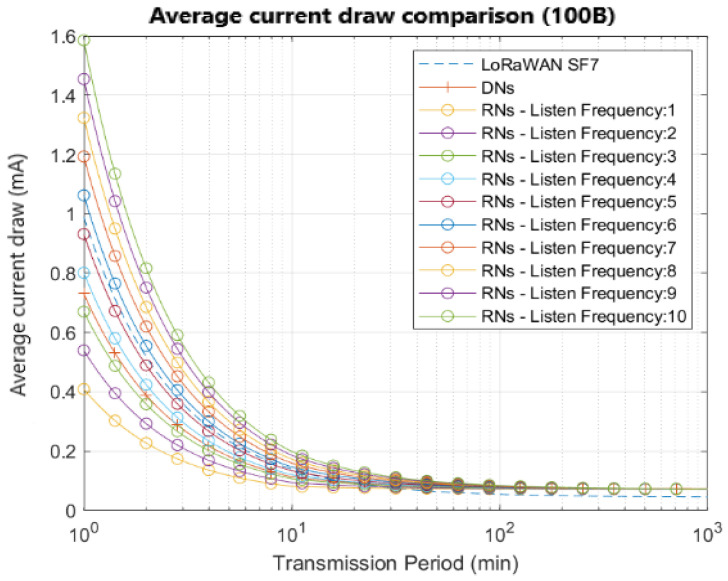
Average current draw in function of transmission period for applications with payloads of 100 bytes.

**Figure 10 sensors-22-02253-f010:**
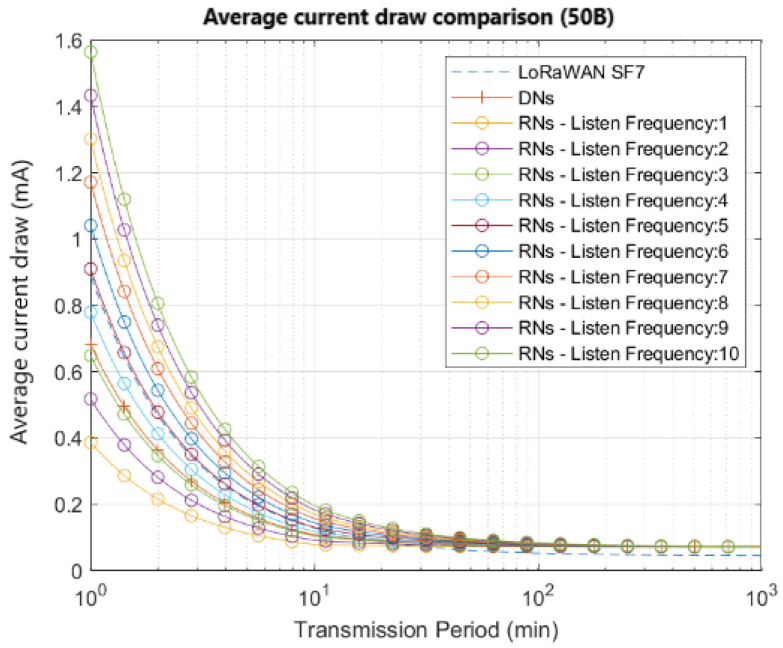
Average current draw in function of transmission period for applications with payloads of 50 bytes.

**Figure 11 sensors-22-02253-f011:**
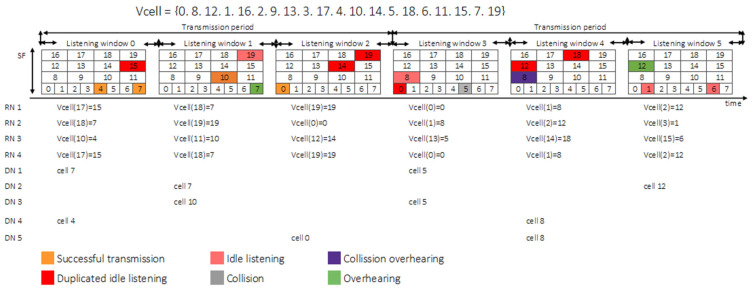
Summary of possible node activity during two transmission periods.

**Figure 12 sensors-22-02253-f012:**
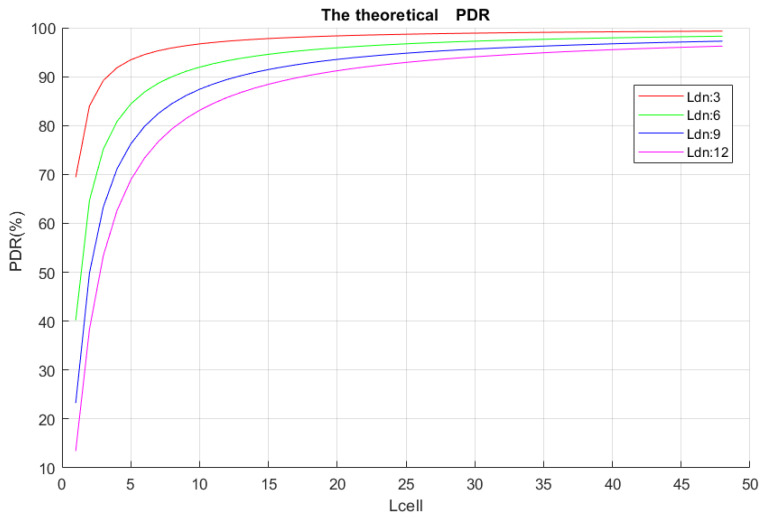
Theoretical performance of the PDR in function of *L_cell_* with *N_LnPerTx_* = 6.

**Figure 13 sensors-22-02253-f013:**
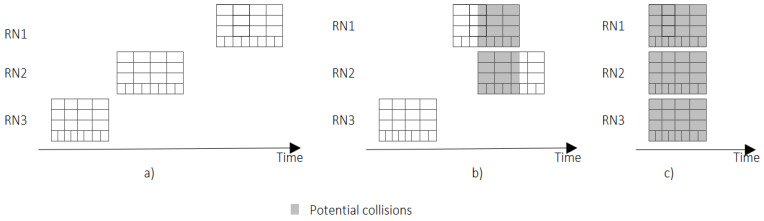
The asynchronous behavior between RNs: (**a**) ideal case, (**b**) typical case, (**c**) worst case.

**Figure 14 sensors-22-02253-f014:**
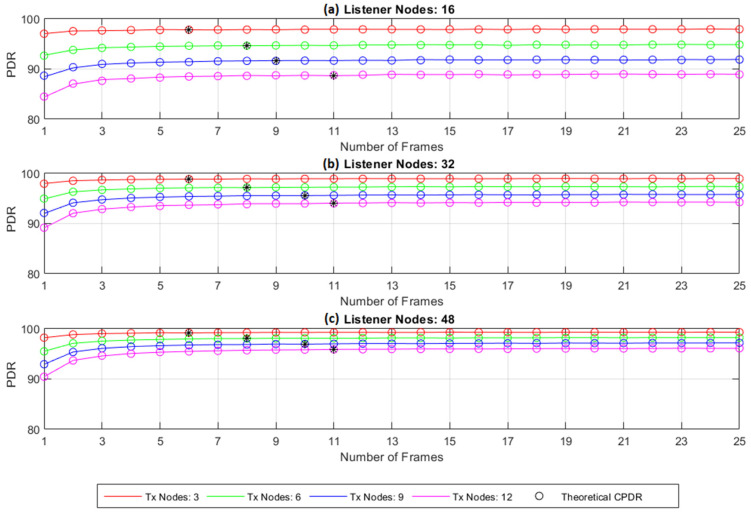
Collision-free packet delivery ratio (PDR) vs. number of frames. (**a**) 16 listener nodes; (**b**) 32 listener nodes; (**c**) 48 listener nodes.

**Figure 15 sensors-22-02253-f015:**
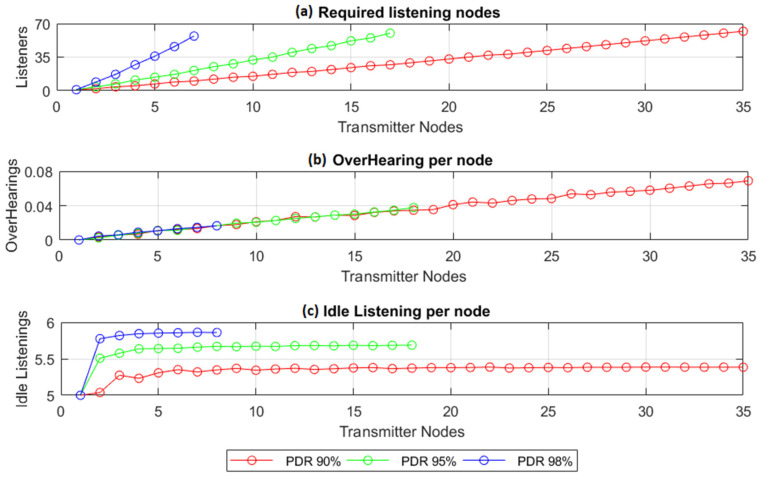
Average performance per number of transmitter nodes in terms of: (**a**) required listening nodes, (**b**) overhearing nodes, (**c**) idle listening per node. The experiment was performed for 50-byte application data using six listening windows in a multi-frame structure of eleven frames of twenty cells each.

**Figure 16 sensors-22-02253-f016:**
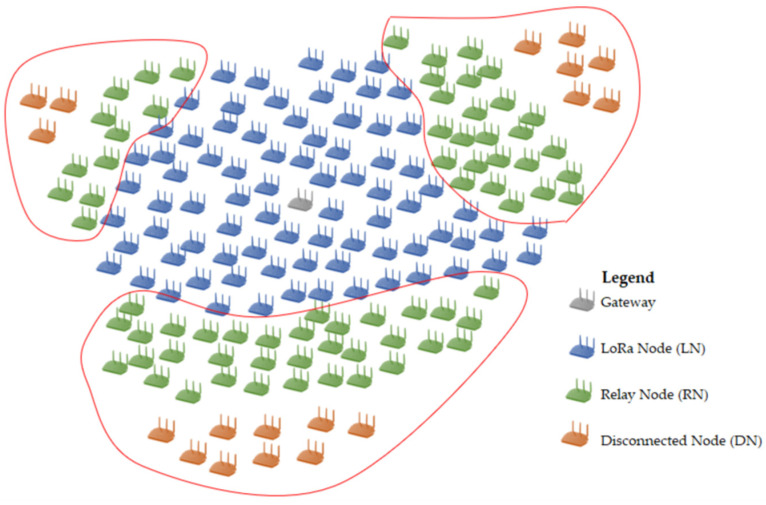
Simulation scenario with three blind spots.

**Figure 17 sensors-22-02253-f017:**
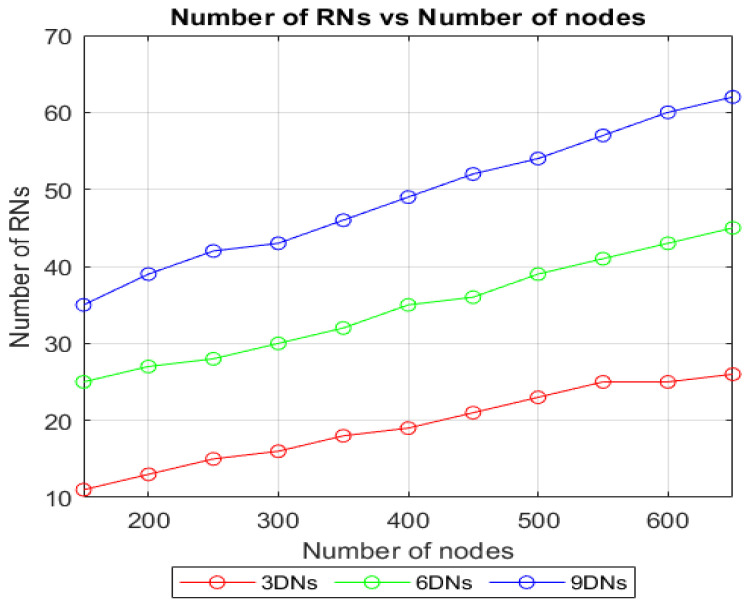
Number of RNs attending each blind spot, increasing with the number of nodes.

**Figure 18 sensors-22-02253-f018:**
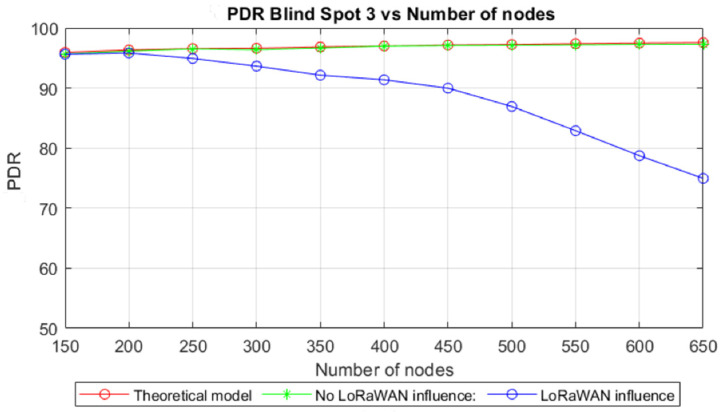
Influence of the LoRaWAN transmissions on the average PDR.

**Table 1 sensors-22-02253-t001:** Summary of previous works.

Ref.	Multi-Hop Supp.	Multi. SF	Energy Analysis	Strong Points	Issues
[8]	yes	no	no	Ensure multi-hop operation to LoRaWAN.	The relay nodes are not energy-constrained devices. Focuses on applications with sporadic traffic. Uses fixed spreading factor.
[9]	yes	yes	no	The use of RPL ensure multi-hop operation.	Energy constraints are not considered or analyzed, and the study is not aimed at high-density networks.
[12]	no	yes	no	Uses multiple SFs.	The transmission opportunities are scheduled by a centralized control. Energy constraints are not considered or analyzed.
[13]	no	yes	no	Increases the scalability and reliability of industrial Internet of things.	Energy constraints are not considered or analyzed.
[14]	yes	yes	no	A successful synchronization for data transmission is ensured by using the proposed time division mechanism.	Does not take advantage of SF orthogonality. The proposal’s impact on energy consumption is not considered.
[15]	yes	no	no	Ensures multi-hop operation.	Uses fixed SF. No energy consumption analysis is carried out.
[17]	yes	no	yes	The proposed synchronous LoRa mesh approach ensures the integration of underground sensors to existing LoRaWAN.	Does not take advantage of SF orthogonality Uses GPS or DCF77 to ensure synchronization. Energy consumption issues are not addressed.
[18]	yes	no	briefly empirically addressed	Ensures multi-hop operation.	Use a fixed SF. A network with low density and low traffic is assumed.
[20]	yes (two hops)	yes	yes (by simulation)	Supports two-hop LoRa network.	Includes hardware modifications.

**Table 2 sensors-22-02253-t002:** Maximum time on air and time-slot duration for TSSFH.

Spreading Factor	ToA (Seconds)	ACK Duration (Seconds)	Computed Time-Slot Duration (Seconds)	Proposed Time-Slot Duration (Seconds)
7	0.3689	0.0617	0.5252	0.6
8	0.6559	0.1132	0.8638	1.2
9	0.6769	0.2058	0.9774	1.2
10	0.6984	0.3707	1.1638	1.2
11	1.4787	0.8233	2.3967	2.4
12	2.7935	1.4828	4.3710	4.8

**Table 3 sensors-22-02253-t003:** States and associated variables, together with their values for the communication process [24].

State Numbers	Description	Duration		Current Draw	
		Variable	Value (ms)	Variable	Value (mA)
1	Wake up	*T_wu_*	168.2	*I_wu_*	22.1
2	Radio preparation	*T_pre_*	83.8	*I_pre_*	13.3
3	Transmit packet/Receive packet/Idle listening	*T_tx_*/*T_rx_*/*T_rx_idle_*	See Table 4	*I_tx_*/*I_rx_*/*I_rx_idle_*	83.0/38.1/38.1
4	Radio transition	*T_switch_*	19.7 [26]	*I_switch_*	13.3
5	Guard time	*T_guard_*	30	*I_guard_*	38.1
6	Transmit ACK/Receive ACK	*T_tx_ACK_*/*T_rx_ACK_*	See Table 4	*I_tx_ACK_*/*I_rx_ACK_*	83.0/38.1
7	Radio off	*T_off_*	147.4	*I_off_*	13.2
8	Postprocessing	*T_post_*	268.0	*I_post_*	21.0
9	Turn-off sequence	*T_seq_*	38.6	*I_seq_*	13.3
10	Sleep	*T_sleep_*	Equation (11)	*I_sleep_*	45 × 10^−2^

**Table 4 sensors-22-02253-t004:** Summary of the different transmission and reception times, based on the overall message time-on-air [27].

DR	SF	*T_tx_/T_rx_* (ms)		*T_tx_ACK_/T_rx_ACK_* (ms)	*T_rx_idle_* (ms)
		50 Bytes	100 Bytes		
2	10	698.4	-	288.7	98.3
3	9	390.1	615.4	144.4	49.15
4	8	215.6	338.4	72.2	24.58
5	7	118.0	189.7	41.2	12.29

**Table 5 sensors-22-02253-t005:** Simulation parameters.

Parameter	Value
Number of runs	100
Frequency band	868 MHz
Number of frequencies	1
Code rate	4/5
Bandwidth	125 kHz
Payload length	50 bytes
Spreading factors	SF7–SF10
Transmission period	15 min
*N_LnPerTx_*	6
*N_p_*	4
Number of frames	11
Cells per frame	20

**Table 6 sensors-22-02253-t006:** PDR comparison for simulation environment and theoretical calculation.

Blind Spot	Theoretical	Simulation
3 DNs vs. 11 RNs	96.92%	96.45%
6 DNs vs. 25 RNs	96.56%	96.20%
9 DNs vs. 35 RNs	95.94%	95.68%

**Table 7 sensors-22-02253-t007:** LoRa WAN traffic influence for blind spots with three and six DNs (values in %).

Number of DNs	Considerations	150	250	350	450	550	650
3 DNs	LoRaWAN influence	96.84	95.99	93.27	90.99	84.24	76.21
	No LoRaWAN influence	96.88	97.59	97.80	98.08	98.52	98.53
	Theoretical model	96.92	97.72	98.09	98.34	98.59	98.65
6 DNs	LoRaWAN influence	96.47	94.92	92.14	89.98	82.89	74.95
	No LoRaWAN influence	96.50	96.56	97.00	97.49	97.56	97.69
	Theoretical model	96.56	96.85	97.24	97.52	97.77	97.97

## Data Availability

Not applicable.

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
