# Peer review of "Time-Slotted Spreading Factor Hopping for Mitigating Blind Spots in LoRa-Based Networks"

_sensors, 2022, doi:10.3390/s22062253_

Round 1

Reviewer 1 Report

Major comments:

  1. The sentence "LoRa radios or motes are suitable in urban areas characterized by harsh propagation environments [2]" on line 35 is not an accurate claim, while it is true that the source claims that, but this literature is outdated and many experiments have been conducted that show that LoRa is not a great fit in urban environments with high interference levels. Please refer to the following:
    1. https://ieeexplore.ieee.org/abstract/document/9395074
    2. https://ieeexplore.ieee.org/document/7925650
    3. https://ieeexplore.ieee.org/document/8903531
  1. In the related work section, please do not only list what has been done but also explain why these methods are not suitable or what are the advantages/disadvantages and add them to Table 1
  2. Using nodes as relays is a very big assumption given that the nodes are really simple/energy constrained. Even if the energy is not of concern, there is a lot of synchronization and transmission repetitions if the transmissions are not received correctly, moreover, interfernece is a big question here because relaying information means congesting the spectrum even further, the authors do not address interference and spectrum access.
  3. A follow up fromt he previous question, what happens when external devices (that are not necessarily LoRa) are using the same spectrum which is the case in the shared spectrum?

Minor comments:

  1. Use larger figures, they are very small and legends cant be seen

Reviewer 2 Report

The authors propose a strategy to improve LoRa network coverage by introducing a two-hop option using a Time Slotted approach. This approach, according to the authors, will provide coverage to locations that were previously without any coverage, or with coverage limited to spreading factors that consumed more energy.

I found the submitted work interesting and tackling the coverage issue by using a novel approach, evaluating it through theorical and simulation of network performance and including an energy consumption evaluation. The English is sound, I only found a bug in "whit" on line 492.

My main suggestion to the authors proposal is the omission of some of the current standardized approaches such as Adaptive Data Rate (ADR) and Listen Before Talk (LBT) to deal with noisy environments. I would prefer to see some discussion in the introduction that would provide the authors view on how ADR and LBT would relate with the work motivation.

Regarding the energy consumption, I think that a review would benefit the work, the authors are presenting what they call energy consumption, but results are measured in mA, something that I would not expect from "energy consumption".

In sum, I find the work interesting, but I think it need a better introduction to motivate the proposal, I would address this by sustaining the work in more updated references and by improving the state of the art. I would also suggest the authors to move to a physical implementation and evaluation, it would increase the proposals' value. The theorical and simulation provided seem to suggest an improvement, however within my own personal experience I doubt that real world results would sustain the authors claims.

Round 2

Reviewer 1 Report

I am happy with the changes. The paper is in good shape.

Author Response

We are grateful to the Reviewer for his/her comments that have been very helpful to improve the quality of the manuscript. In the revised manuscript we make a language and spelling check and modify few small mistakes.

Reviewer 2 Report

Dear Authors,

Thank you for your revised version.

I think all of my concerns are fully addressed. I would only suggest authors to clarify the newly introduced statement on line 69 about the duty cycle limit of 1%. This is true if we are only talking about end devices in LoRaWAN. LoRaWAN Gateways have the RX2 downlink channel which uses a frequency where a duty cycle of 10% is possible, assuming a network planning according to the default values. When writing about broad "LoRa Devices" stating 1% as a limitation is not totally correct, since it depends on the used channel.

Author Response

We are grateful to the Reviewer for his/her comments that have been very helpful to improve the quality of the manuscript. 

Reviewer comment: I think all of my concerns are fully addressed. I would only suggest authors to clarify the newly introduced statement on line 69 about the duty cycle limit of 1%. This is true if we are only talking about end devices in LoRaWAN. LoRaWAN Gateways have the RX2 downlink channel which uses a frequency where a duty cycle of 10% is possible, assuming a network planning according to the default values. When writing about broad "LoRa Devices" stating 1% as a limitation is not totally correct, since it depends on the used channel.

Answer: Thanks a lot for your comment. We recognize that the original statement on line 69 about the duty cycle limit of 1% may have not been sufficiently clear so, in the revised manuscript we modify the sentence in lines 68-70 as “The TSSFH spectrum access method restricts the duty cycle to 1%, a mandatory legal restriction for uplink communication of LoRaWAN end devices that use the 868 MHz ISM band in Europe

Additionally we make a language and spelling check and modify few small mistakes.